# Product Evaluation Prediction Model Based on Multi-Level Deep Feature Fusion

**Qingyan Zhou** [1,*], **Hao Li** [1], **Youhua Zhang** [1] **and Junhong Zheng** [2]

1   School of Information and Computer, Anhui Agricultural University, Hefei 230036, China
2   School of Computer Science and Technology, Zhejiang Sci-Tech University, Hangzhou 310018, China
*   Correspondence: qingyanzhou@ahau.edu.cn

**Abstract:** Traditional product evaluation research is to collect data through questionnaires or interviews to optimize product design, but the whole process takes a long time to deploy and cannot fully reflect the market situation. Aiming at this problem, we propose a product evaluation prediction model based on multi-level deep feature fusion of online reviews. It mines product satisfaction from the massive reviews published by users on e-commerce websites, and uses this model to analyze the relationship between design attributes and customer satisfaction, design products based on customer satisfaction. Our proposed model can be divided into the following four parts: First, the DSCNN (Depthwise Separable Convolutions) layer and pooling layer are used to combine extracting shallow features from the primordial data. Secondly, CBAM (Convolutional Block Attention Module) is used to realize the dimension separation of features, enhance the expressive ability of key features in the two dimensions of space and channel, and suppress the influence of redundant information. Thirdly, BiLSTM (Bidirectional Long Short-Term Memory) is used to overcome the complexity and nonlinearity of product evaluation prediction, output the predicted result through the fully connected layer. Finally, using the global optimization capability of the genetic algorithm, the hyperparameter optimization of the model constructed above is carried out. The final forecasting model consists of a series of decision rules that avoid model redundancy and achieve the best forecasting effect. It has been verified that the method proposed in this paper is better than the above-mentioned models in five evaluation indicators such as MSE, MAE, RMSE, MAPE and SMAPE, compared with Support Vector Regression (SVR), DSCNN, BiLSTM and DSCNN-BiLSTM. By predicting customer emotional satisfaction, it can provide accurate decision-making suggestions for enterprises to design new products.

**Keywords:** product evaluation prediction; Depthwise Separable Convolutions; Bidirectional Long Short-Term Memories; attention mechanism; genetic algorithm

## 1. Introduction

With the development of mobile Internet and e-commerce, after customers buy products online and experience them, they will generally write down their experience of using the products on the platform. Research on how to efficiently mine the emotional preferences of consumers to publish online reviews can provide consumers with scientific purchasing decision guidance, provide enterprises with targeted production design methods, improve customer experience and increase potential consumer groups [1]. However, in the face of such a large amount of comment data, it is not possible to control the market's diversified needs for products only from the perspective of enterprises. In Quality Function Deployment (QFD), The House of Quality (HOQ) is used to identify the link between customer requirements and the performance of the corresponding product or service to help product designers determine the optimal design attribute settings [2]. However, due to the need for a large number of questionnaires, the formulation of the HOQ is a long process, and after a long period of processing, the customer-oriented product design cannot be carried out

in a timely and accurate manner. The new method is to extract key words and calculate the similarity of words in online reviews, parameterize the product image on the research object, construct the mapping relationship between product feature parameters and image parameters using BP (Back Propagation) neural network to form the product review model, and guide the design through the evaluation results [3]. The authors in [4] proposed a method. First, the semi-supervised learning recursive autoencoder (ss-rae) model is used to classify the comments of electric water heaters on the sentiment tendency, and the customer satisfaction is obtained according to the sentiment classification results, predict customer satisfaction over the next unit time. Determining the design attributes of new products is crucial to maximizing customer satisfaction. This research [5] proposed a data-driven product optimization design method, using the K-means algorithm to perform cluster analysis on rice cooker reviews, and optimize according to the clustering results. After encoding the target features, the Non-inferior Sorting Genetic Algorithm II (NSGA-II) was used to optimize the product features, and a rice cooker oriented to customer preferences was designed. This research [6] proposed a genetic algorithm constrained by bidirectional association rules to predict customer satisfaction and provide product design suggestions for enterprises.

To sum up, the traditional product design plan that considers customer preferences uses the SD (Semantic Differential) questionnaire method to collect data, and then establishes a model through experience to obtain the design plan, which is time-consuming and labor-intensive. Recent research mainly uses online reviews to quantify product imagery, and establishes a model based on the method of Kansei Engineering to analyze the mapping relationship between product experience information and product design elements, so as to carry out customer-oriented product optimization design [7–9]. For the complex relationship between design elements and customer satisfaction, the existing research is not accurate enough to predict customer evaluation satisfaction, and a more efficient model is needed to predict customer evaluation satisfaction from the perspective of multiple design elements [10,11], such as product color, The impact of quality and performance on customer satisfaction, and then provide more scientific decision-making for production design. Therefore, this paper proposes a multi-level deep feature fusion product evaluation prediction model, which has the following salient features in product evaluation prediction:

- The model uses the fine-grained emotional quantification of various evaluations of consumers' online shopping products, and obtains the emotional value sequence data of the comprehensive customer experience.
- DSCNN (Depthwise Separable Convolutions) to extract primary interconnected features from the data set affecting customer satisfaction, use CBAM (Convolutional Block Attention Module) to implement multi-dimensional separation feature attention strategy in channel and space dimensions, and analyze shallow feature channels and spatial dimensions through channel attention mechanism The importance of the multi-features is assigned different weights by different importances, which avoids the problem of poor prediction effect caused by the loss of important features during the training of the multi-hidden layer model.
- Optimize BiLSTM (Bidirectional Long Short-Term Memory) learning performance by CBAM attention mechanism. BiLSTM is an improvement of RNN (Recurrent Neural Network) and LSTM (Long Short-Term Memory) [12–14]. In order to solve the problem of gradient disappearance and gradient explosion, it can process the combined features of the front and rear bidirectional sequences at the same time, thereby obtaining the feature map of the secondary deep analysis of the data.
- Construct a multi-level deep feature fusion consisting of a channel-by-channel convolution layer, a point-by-point convolution layer, a maximum pooling layer, a weight distribution layer for channel and spatial dimension features, a bidirectional LSTM prediction layer, and a multi-layer Dense output layer. Product evaluation prediction model. The global optimization of the multi-layer model structure is carried out through the genetic algorithm, which highlights the learning advantages of each layer

and eliminates the barriers of independent learning between each layer [15,16]. The ultimate goal is to improve the prediction accuracy of product evaluation satisfaction.

## 2. Using Genetic Algorithm to Optimize Spatiotemporal Correlation Forecast Model

Since the classical prediction algorithm can only extract shallow features, it lacks the analysis of deep-level information for multivariate feature prediction, resulting in low long-term prediction accuracy. This paper Degree of satisfaction a spatiotemporal correlation (DSCNN-CBAM-BiLSTM) prediction model that integrates the attention mechanism (as shown in Figure 1). Firstly, the DSCNN is used to replace the classical convolution to extract the multi-dimensional and multi-feature information of the data, and then the attention mechanism of the convolution module is added. Combined with the attention mechanism, the learning performance of BiLSTM network is optimized, and the degree of attention to local important information in the sequence after one feature extraction is strengthened. Finally, the bidirectional LSTM is used to realize forward and backward bidirectional analysis of sequence data, realize the extraction of secondary features, and further solve the problems of missing important features and excessive model hyperparameters, thereby greatly improving the accuracy of prediction.

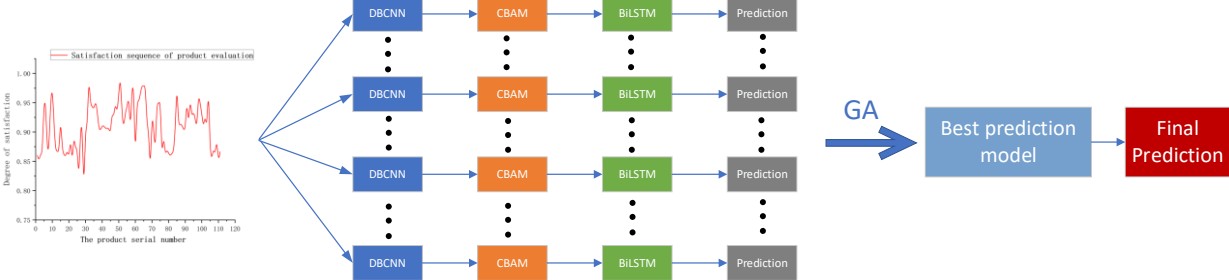

**Figure 1.** Using Genetic Algorithm (GA) to optimize Spatiotemporal correlation forecast model.

### 2.1. Spatiotemporal Correlation Prediction Model

In the existing literature, there are few researches on sentiment index of product evaluation, which is a very challenging work, because consumers' online reviews of products after purchase are full of complex emotions. If we set aside some subjective factors, consumer satisfaction with a product mainly depends on the quality and characteristics of the product itself. For example, for monitors, crawling reviews and analyzing high-frequency words revealed that the price, brand, resolution, panel material, weight and screen size of the monitor were the main factors influencing consumer reviews. Multivariable prediction problem is a complex problem. Therefore, we use DSCNN-BiLSTM prediction model with attention mechanism to analyze sequence data from multiple dimensions. The goal is to improve the generalization ability of the model, we also use genetic algorithm to optimize the structure of the model to ensure the accuracy of prediction from multiple aspects.

#### 2.1.1. The First Feature Extraction Based on DSCNN

The classic Convolutional Neural Network (CNN) is a deep learning model proposed by Yan LeCun in 1998 and applied to the field of computer vision. It is to achieve dimensionality reduction of high-dimensional data while performing feature extraction through convolution and pooling operations. Due to the excessive number of parameters and the increase of computational cost in the process of feature extraction by classical CNN, to solve this problem, Xception [17] separates spatial convolution and channel convolution, and separates the correlation in two directions. MobileNet [18] adopted DSCNN to optimize the model complexity while improving performance. In the model of this paper, the classical convolution is replaced by the DSCNN operation, which aims to achieve a shallow global feature extraction from the two dimensions of space and channel.

Figure 2 shows the structure of the DSCNN, the input feature map size is C × H. It is mainly optimized on the network structure and consists of two processes: Depthwise

Convolution (DWC) and Pointwise Convolution (PWC). This approach not only conforms to the law of multi-dimensional feature extraction, but also can freely adjust the model representation ability compared with standard convolution.

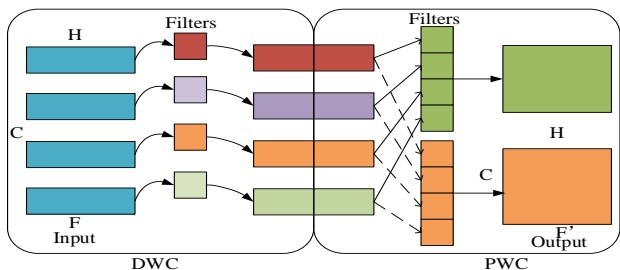

**Figure 2.** DSCNN structure diagram.

### 2.1.2. The Second Feature Extraction Based on BiLSTM

LSTM is an improved result of the traditional Recurrent Neural Network (RNN), which can perform long-term analysis and prediction on sequence data and effectively prevent the gradient from disappearing during RNN training. The LSTM structure (as shown in Figure 3) adopts the control gate mechanism, which is composed of memory cells, input gates, output gates, and forgetting gates. Among them, $x_t$ is the input of the current time, and $h_t$ is the state of the cell.

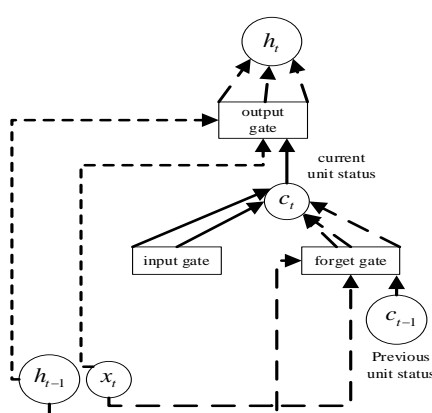

**Figure 3.** LSTM structure diagram.

The input gate $i_t$ receives the current input $x_t$ and the final hidden state $h_{t-1}$ as input, and calculates $i_t$ according to Formula (1).

$$i_t = \sigma(W_{ix}x_t + W_{ih}h_{t-1} + b_i) \tag{1}$$

After calculation, when $i_t$ is 0, it means that any information input will not enter the unit state, and when $i_t$ is 1, it means that all the information currently input will enter the unit state. In Formula (2), $c_t{}^\sim$ will calculate another value, called candidate value, which is used to calculate the current cell state.

$$c_t{}^\sim = \tan h(W_{cx}x_t + W_{ch}h_{t-1} + b_c) \tag{2}$$

The forget gate will do the following: a forget gate value of 0 means that no information about $c_{t-1}$ is passed to the computation of $c_t$, and a value of 1 means that all information is passed to $c_t$.

$$f_t = \sigma(W_{fx}x_t + W_{fh}h_{t-1} + b_f) \tag{3}$$

The final $h_t$ state of the LSTM cell:

$$o_t = \sigma(W_{ox}x_t + W_{oh}h_{t-1} + b_o) \tag{4}$$

$$h_t = o_t \tan h(c_t) \tag{5}$$

Equations (1)–(5), $W_{ix}$, $W_{ih}$, $W_{cx}$, $W_{ch}$, $W_{fx}$, $W_{fh}$, $W_{ox}$, $W_{oh}$ represent the weight matrix of each control gate; and $b_i$, $b_c$, $b_f$, $b_o$ represent the bias of each control gate; σ and tanh are the sigmoid and tanh activation functions, respectively.

BiLSTM is composed of two LSTMs with the same structure and opposite directions (as shown in Figure 4), so it can process satisfaction sequence of product evaluation in both directions at the same time, better capture the dependencies between features, fully mine the hidden information in the data, and make overall plans. Considering factors such as historical evaluation satisfaction and price, secondary correlation features can be extracted to further improve the prediction accuracy [19].

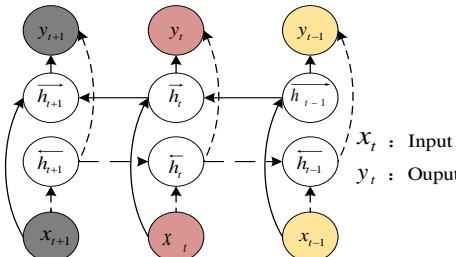

**Figure 4.** BiLSTM structure diagram.

## 2.2. Multi-Channel and Multi-Location Attention Mechanism

In the DSCNN-BiLSTM prediction model, the information contained in each channel and spatial position is regarded as equally important due to the classical convolution operation. If the weights of important features and general features are the same, the accuracy of the model prediction will be greatly reduced [20,21]. The attention mechanism can focus more on the information that has a greater effect on the current output results, reduce the attention to redundant information, and even filter out irrelevant information under the condition of limited computing power. This research [22] proposed an improved lightweight convolutional attention module (CBAM). It contains two independent sub-modules, namely Channel Attention Module (CAM) and Spatial Attention Module (SAM), realizes the attention mechanism separated in channel and spatial dimensions (as shown in Figure 5), and also proves that the specific combination of CAM and SAM in the form of CAM followed by serial SAM, the effect will be better. The calculation formula of CBAM can be expressed as Formula (6) and Formula (7), where the input feature map is denoted as F, and ⊗ means element-wise multiplication [23,24].

$$F' = Mc(F) \otimes F \tag{6}$$

$$F'' = Ms(F') \otimes F' \tag{7}$$

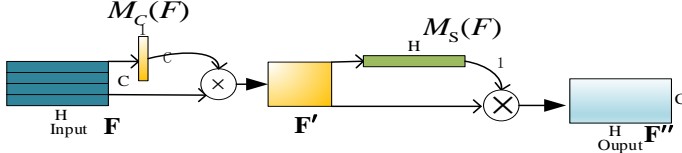

**Figure 5.** CBAM network structure diagram.

### 2.2.1. Attention Module of Channel Dimension

The main idea of CAM is to perform a global pooling operation in the channel dimension, obtain the weights through the shared network output, and then add them as the final attention vector. The specific structure of CAM is shown in Figure 6, and the main process can be expressed by Equation (8). In this paper, the dimension of *F* is $7 \times 4$, and the feature map of size $4 \times 55$ is obtained after the convolution pooling operation. AvgPool(F) and MaxPool(F) represent the average pooling feature and max pooling feature operations,

respectively, and σ is the sigmoid function. After the operations of $F_{avg}^c$ and $F_{max}^c$, two corresponding feature maps are generated. The shared network consists of a Multi-layer Perceptron (MLP), which has a hidden layer, and W0 and W1 are the two weights of the MLP. After applying the shared network to each feature map, the output feature vectors are summed and subjected to the sigmoid activation function to obtain the channel attention feature Mc(F). After Mc(F) is multiplied by the input feature map F, the feature map F′ with different weights on each channel is obtained.

$$
\begin{aligned}
Mc(F) &= \sigma(MLP(AvgPool(F)) + MLP(MaxPool(F))) \\
&= \sigma\left(W1\left(W0(F_{avg}^c)\right) + W1(W0(F_{max}^c))\right)
\end{aligned}
\tag{8}
$$

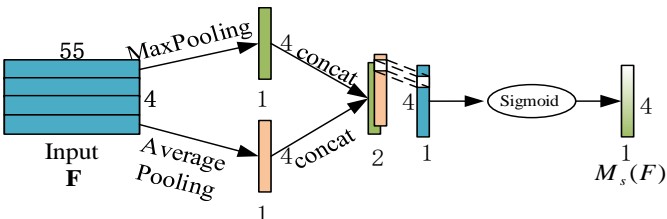

**Figure 6.** CAM structure diagram.

2.2.2. Attention Module in Spatial Dimension

　　The structure of the spatial attention module (SAM) is shown in Figure 7. First of all, the global maximum pooling and maximum average pooling operations are performed on the input feature map to obtain two sub-feature maps, which are aggregated into a 2-channel feature map by channel. They are then connected through standard convolutional layers and subjected to convolution operations to generate a spatial attention feature submap Ms(F) with a channel size of $1 \times 4$. The calculation process of SAM in this model can be expressed as Equation (9).

$$
\begin{aligned}
Ms(F) &= \sigma\left(f^7([AvgPool(F); MaxPool(F)])\right) \\
&= \sigma\left(f^7\left(\left[F_{avg}^s; F_{max}^s\right]\right)\right)
\end{aligned}
\tag{9}
$$

where σ refers to the sigmoid function, and $f^7$ represents a one-dimensional convolution operation with a filter size of 7. After the sigmoid activation function, the spatial attention feature map Ms(F) is obtained. Ms(F) is multiplied by the feature map F′ generated by channel attention, and the final feature map F″ is output.

**Figure 7.** SAM structure diagram.

## 3. Model Parameter Optimization Based on Genetic Algorithm

　　Due to the prediction model of BiLSTM, DSCNN and Dense layers proposed in this paper, too many adjustable parameters make the problem very complicated. This complexity is not only related to the form of the incoming data at each layer, but also to the choice of the precise structure of the neural network. Consequently, we use the GA to optimize the model structure (as shown in Figure 8), which is evolved according to the theory of biological evolution in nature [25–30]. Following work needs to be done to ensure the normal execution of the genetic algorithm.

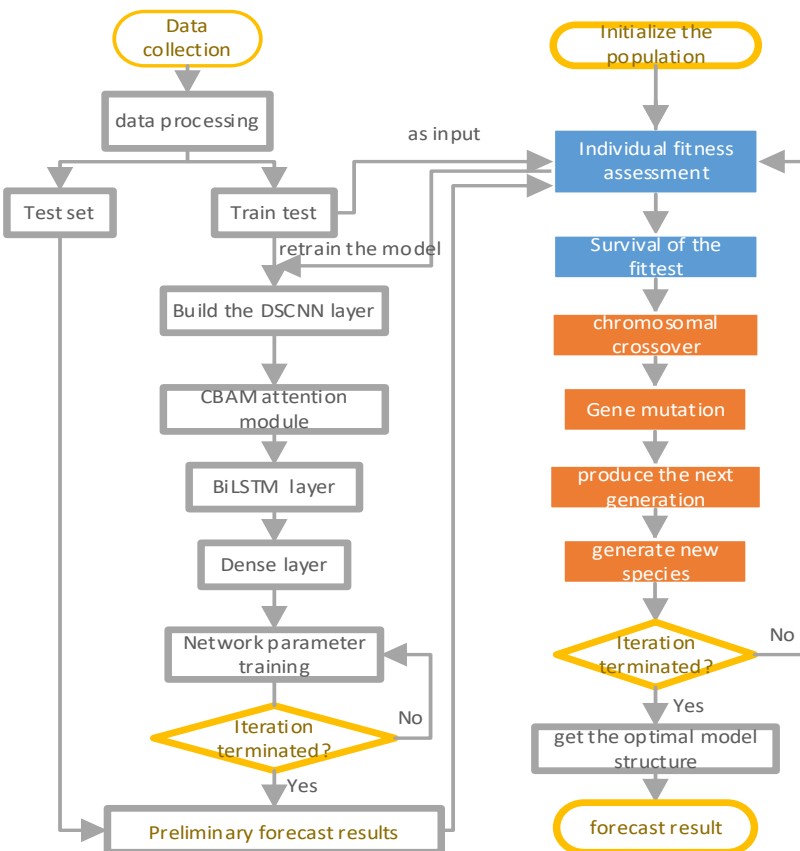

**Figure 8.** Genetic Algorithm optimization flow chart.

1. First, determine the model structure that needs to be optimized, including the number of deep convolutional layers, the number of BiLSTM layers, the number of fully connected layers, and the number of neurons in each of the above layers.
2. Secondly, set the parameters of the genetic algorithm (as shown in Table 1), where population represents the number of individuals in the population, $n_c$ is the probability of parental chromosome gene recombination, $n_m$ is the probability of gene mutation, and $E_p$ represents the evolutionary generation of the genetic algorithm.

**Table 1.** Genetic algorithm parameter list.

| Parameter | Value |
|-----------|-------|
| population | 10 |
| $n_c$ | 0.5 |
| $n_m$ | 0.2 |
| $E_p$ | 20 |

3. The next step is to create the first population of the neural network, each of which is initialized with a random combination of descriptive attributes. In this paper, parameters to be optimized are put into the list and encoded with integers. There are 12 attributes to be optimized in the model, so the chromosomes of individuals in the population can be instantiated into a list containing 12 integers. When the above coding is completed, an individual genotype is formed, which is a possible value for the optimal solution of the model. The size of population in the genetic algorithm represents the number of individuals in the population. Random value is used to randomly assign value to each chromosome of all individuals. At this time, there are 10 types of individuals in the first-generation population, namely, there are 10 models with different structures, and each individual represents a possible optimal solution.

4.  After initialising the first generation population, the next step is to train the ten models using the dataset in this paper. The loss function during training is the mean square error (MSE) and the optimiser is Adam, with the goal of minimising the MSE. Other metrics used to evaluate the performance of the models were used to evaluate the performance of the models on the test set after training was completed, with one of the performance metrics used as a function of the fitness of the genetic algorithm to evaluate the merit of the models.

5.  When all individuals in the population are evaluated, the higher the fitness, the greater the probability of being selected for retention as a parent. The parent generation left from the previous generation is crossed by two chromosomes with $n_c$ probability, and gene mutation and generation of the next generation occur with $n_m$ probability, finally forming a new population. When twenty populations are generated, the iteration of genetic algorithm ends, and the individuals with the highest fitness are selected as the optimal model structure, so as to output the prediction results.

The structure of the model optimized by GA in this paper is shown in Figure 9. In Figure 9, 1 represents the DSCNN,2 represents the CBAM attention mechanism, 3 represents the BiLSTM layer, and 4 represents the fully connected layer. The final output is made by the fully connected layer. It shows the model structure with the best prediction performance after optimization by GA, including the input feature map size, the parameters of DSCNN, BiLSTM and the number of stacked layers. The role of the genetic algorithm is to eliminate the poorly performing models and retain the optimal model.

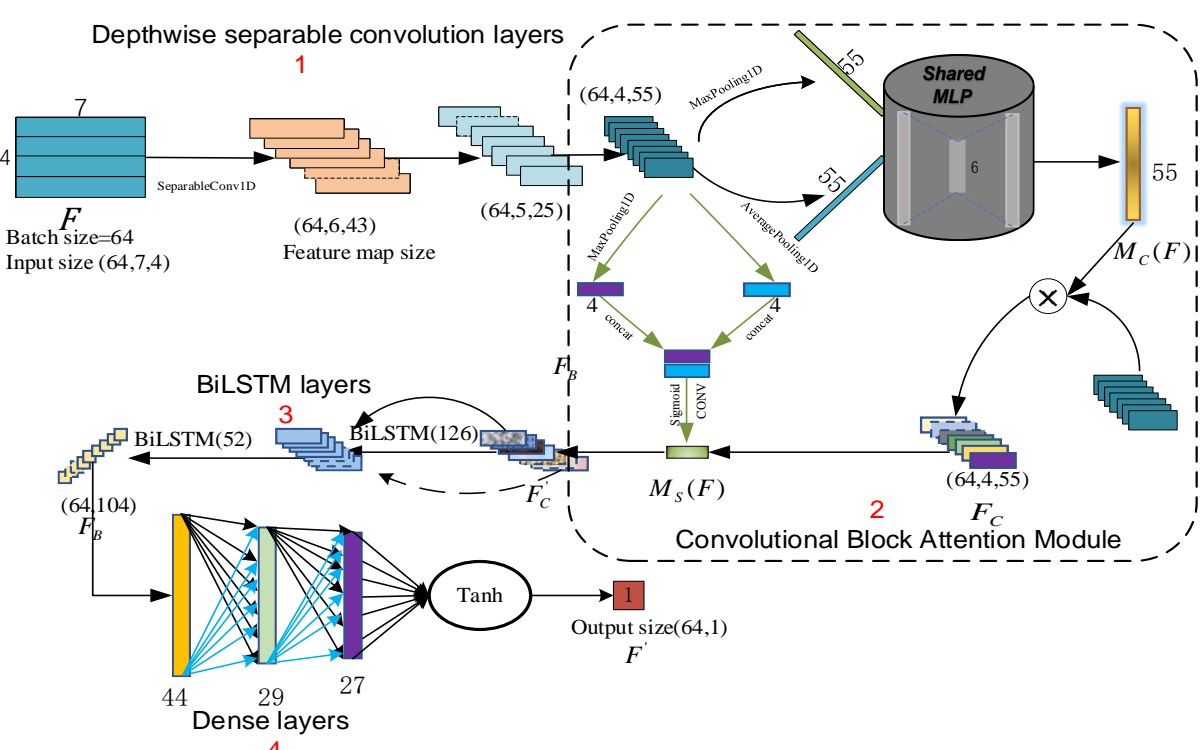

**Figure 9.** Spatiotemporal correlation prediction model with attention mechanism.

## 4. Experiment and Result Analysis

In this section, customers' satisfaction with product evaluation is obtained through online comments, and product parameter data affecting satisfaction is extracted to form the data in this paper, and the optimal network structure obtained by genetic algorithm is used for prediction. In order to verify and evaluate the prediction results of the model, five model evaluation indicators were evaluated, and compared with classical prediction algorithms such as BiLSTM and Support Vector Regression (SVR), The results show that our model has a high prediction accuracy.

In addition, the experimental environment of this paper is as follows: Python3.8, Tensorflow-gpu2.4.0, keras2.4.3. The computer configuration is CPU: R5-5600h, GPU: 3050Ti, RAM: 16G.

### 4.1. Experimental Data

By crawling the basic information and comments of multiple displays on an e-commerce website, six attribute parameters of price, brand, screen resolution, panel material, weight and screen size of the display are selected manually and comprehensively from the collected data, and these attributes are regarded as input variables affecting customer satisfaction with the product. The post-purchase reviews of consumers are converted into customer satisfaction of the product as an output variable. After that, the API interface of sentiment tendency analysis in Baidu natural language processing was called, and each comment was input to calculate the sentiment tendency index (as shown in Table 2). The average value of sentiment evaluation of each product was taken as the customer satisfaction λ of the display to construct the sentiment value sequence data. The range of emotion value is [0,1], where 1 represents excellent evaluation and 0 represents extreme poor. In addition, our original data contains text and numerical values, which are converted into pure numerical types through feature engineering and normalized to improve the efficiency of model training.

**Table 2.** Sentiment analysis of product evaluation.

| Comment | λ |
|---|---|
| The effect is super, the connection setting is convenient, go back and test the refresh rate and rgb color gamut. | 0.999909 |
| The power interface is easy to loosen, and the system response speed is slow. | 0.000644 |
| The screen is okay, and that's it for the price. | 0.809492 |

The original data is digitally encoded, and the digitally encoded data is normalized (as shown in Equation (10)). $x_i$ represents the normalized sampling data, $x_{max}$ and $x_{min}$ represent the maximum and minimum values in the normalized features. The processing flow of the experimental data is shown in Figure 10. Our experiment contained 2560 pieces of data, of which 80% was used as the training set, 10% as the validation set, and the remaining 10% as the test set. The original data feature number is 7, the sliding window is set as 4, and the batch size is set as 64, so the dimension of the input composite sequence feature data is (64,7,4).

$$x_i^* = \frac{x_i - x_{min}}{x_{max} - x_{min}} \tag{10}$$

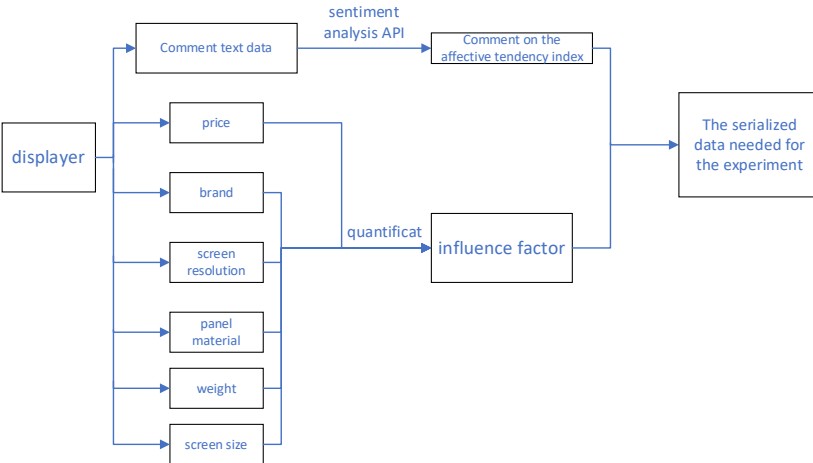

**Figure 10.** Flowchart of the transformation of raw data into serialized data.

### 4.2. Model Optimization Experiment

When using genetic algorithm to optimize model structures, in addition to determining the parameters in Table 1 above, it is also important to choose the appropriate fitness

function and the number of iterations of the predictive model, which directly affect the speed of convergence of the algorithm and the ability to find an optimal solution.

### 4.2.1. Model Performance Evaluation Indicators

To evaluate the prediction effect of the model, MAE, MSE, MAPE, RMSE and SMAPE are selected as the performance evaluation indicators of the model in this paper, and the calculation formulae are shown in Equations (11)–(15) respectively.

$$\mathrm{MAE} = \frac{1}{n} \sum_{i=1}^{n} |\hat{y}_i - y_i| \tag{11}$$

$$\mathrm{MSE} = \frac{1}{n} \sum_{i=1}^{n} (\hat{y}_i - y_i)^2 \tag{12}$$

$$\mathrm{MAPE} = \frac{1}{n} \sum_{i=1}^{n} \left| \frac{\hat{y}_i - y_i}{y_i} \right| \times \%100 \tag{13}$$

$$\mathrm{RMSE} = \sqrt{\frac{1}{n} \sum_{i=1}^{n} (\hat{y}_i - y_i)^2} \tag{14}$$

$$\mathrm{SMAPE} = \frac{1}{n} \sum_{i=1}^{n} \frac{2|\hat{y}_i - y_i|}{|\hat{y}_i| + |y_i|} \times \%100 \tag{15}$$

In Equations (11)–(15), $n$ is the total number of test samples, $y_i$ represents the true value, $\hat{y}_i$ is the predicted value, and $\overline{y}$ is the mean value of the true value. The smaller the MAE, MSE, MAPE, RMSE and SMAPE indicators, the better the model performance. MAE, MSE, MAPE and RMSE were selected as fitness function of genetic algorithm for comparative experimental study. In this study, the fitness should be normalized first, and the fitness value is mapped to [0.01,0.99]. Equation (16) reflects the probability $P_i$ of individual selection.

$$P_i = \frac{1 - f_i^*}{\sum_{i=0}^{n}(1 - f_i^*)} \times \%100 \tag{16}$$

In the Equation (16): $f_i^*$ represents the fitness value after mapping, and n represents the total number of individuals in the population.

### 4.2.2. Model Hyperparameter Tuning

Once a prediction model is established, the parameters to be determined are the model's epoch. If the epoch is too large, it will easily cause overfitting. If the epoch is too small, it will cause overfitting, which will affect the fitness calculation of genetic algorithm. Therefore, this paper selects a group of epoch values of 10, 50, 100 and 150 to carry out experiments respectively. The results show that when epoch is equal to 100, the genetic algorithm has the best effect. After that, we take MSE, MAE, RMSE and MAPE as fitness functions, when epoch is set to 100 and MAE, MSE and RMSE are used as fitness functions, the fitness function image obtained by genetic algorithm is shown in Figure 11.

The choice of an appropriate genetic function directly affects the speed of convergence of the model and the ease of finding an optimal solution. Therefore, different performance metrics are chosen as the fitness function for comparison experiments. It can be concluded from Table 3 that the prediction effect of the model was the best when MAPE was used as the objective function. Compared with MSE as fitness function, the model in this paper improved by %1.10 and 0.59% on MAPE and SMAPE respectively. Therefore, MAPE index was selected as the fitness function of the model in this paper. Chromosome 9 of the 20th generation has the highest fitness, which can be expressed as [3,2,3,43,25,55,63,52,44,29,27,0] in the form of a list. The first three loci represent the number of DSCNN layers, BiLSTM layers and Dense layers, respectively. The subsequent loci represent the number of neurons on each layer, thus obtaining the final model structure.

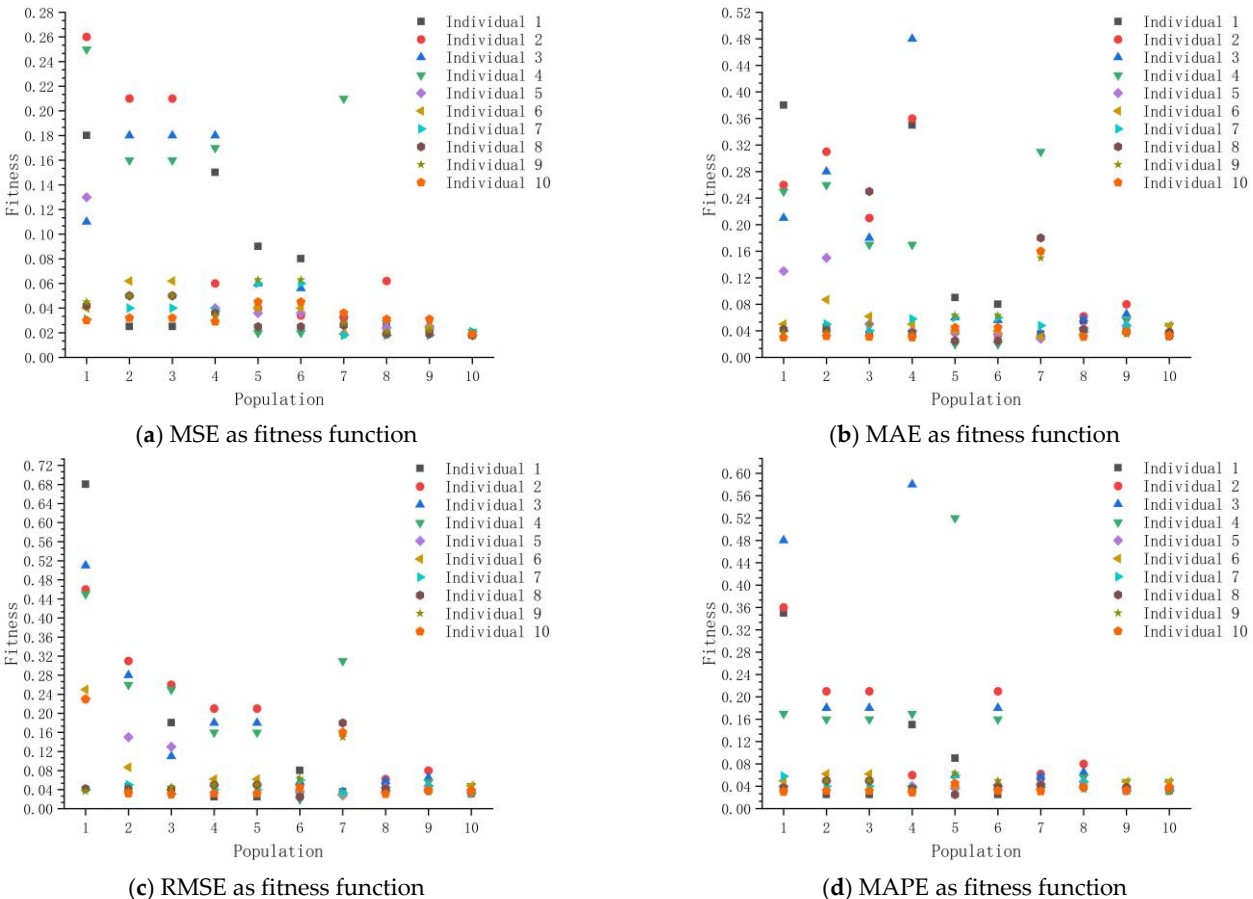

**Figure 11.** Fitness graph under different fitness functions.

**Table 3.** Model performance under different fitness functions.

| Fitness Function | MAE | MSE | RMSE | MAPE | SMAPE |
|---|---|---|---|---|---|
| MAE | 0.0322 | 0.0015 | 0.0389 | %3.5460 | %3.5496 |
| MSE | 0.0370 | 0.0018 | 0.0420 | %4.0497 | %4.0714 |
| RMSE | 0.0326 | 0.0015 | 0.0388 | %3.5929 | %3.5840 |
| MAPE | 0.0269 | 0.0010 | 0.0316 | %2.9543 | %2.9597 |

*4.3. Comparative Experiment*

SVR were used to conduct prediction experiments on the data in this paper, and the prediction results of each model were visualized (as shown in Figure 12). SVR is widely used to predict time series and multivariable prediction problems.

In this section, the ablation experiment is carried out. DSCNN-BiLSTM and the model in this paper are combined with DSCNN and BiLSTM as the benchmark model, and Adam optimizer is used for training of all models. Adam optimizer parameters are set as follows: The learning rate = 0.001, $\beta_1$ = 0.9, $\beta_2$ = 0.009, the loss function is MSE, and the dropout is set to 0.3. The convolution kernel size of DSCNN, DSCNN-BiLSTM and the convolution layer of the model in this paper was set as 2, and RELU was used as the activation function to reduce the amount of calculation and prevent over-fitting. The above model each layer after adding batch of standardized BN, in the training process converts the distribution of each layer of neurons input values to the normal distribution, reduce internal covariance deviation, gradient bigger to avoid gradient disappeared, to accelerate the convergence speed, enhances the model generalization ability, the dropout technology in the whole connection layer to join L2 regularization operation, overfitting is further prevented.

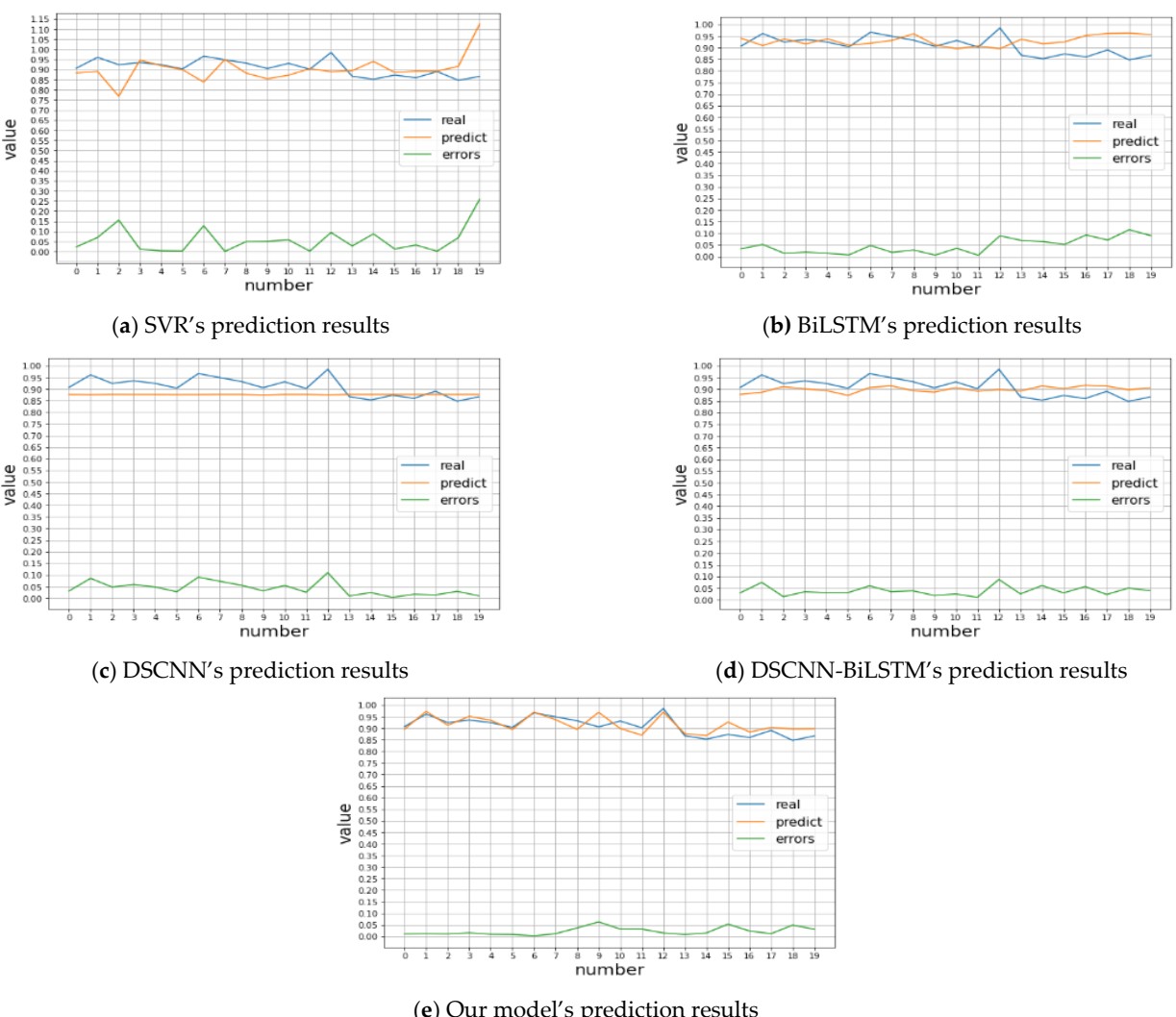

(**a**) SVR's prediction results

(**b**) BiLSTM's prediction results

(**c**) DSCNN's prediction results

(**d**) DSCNN-BiLSTM's prediction results

(**e**) Our model's prediction results

**Figure 12.** Graph of predicted results of different models.

Figure 12 shows the predicted results of ablation experiments conducted with the model in BiLSTM, DSCNN and DSCNN-BiLSTM respectively. Number represents the product number in the validation data, represents a total of 20 displays, and Value represents the satisfaction Value of customer evaluation. It can be clearly seen from the figure that the prediction curve of machine learning SVR model deviates significantly from the actual value, and the absolute prediction error is obviously greater than 0, indicating poor prediction effect. However, DSCNN, BiLSTM benchmark model prediction model and its combination prediction model using deep learning have improved the prediction effect, but compared with ARIMA with accurate modeling, it is easy to fall into local extremes, leading to problems such as overfitting, which makes it difficult to achieve the best prediction results. In this paper, the attention module is embedded in the DSCNN-BILSTM and the model structure is globally optimised using genetic algorithms. Compared with the other five models, the prediction curve of this model is closer to the true value curve and the absolute error is closer to zero.

The prediction results of the models in this paper were compared with those of the other models (as shown in Table 4). In the evaluation and prediction of 20 displays, the prediction effectiveness of this paper's model was improved to varying degrees compared to the other five models. Compared with the machine learning SVR model, the prediction effectiveness of this paper's model was reduced by about 3.35% and 3.29% for MAPE and SMAPE respectively. To some extent, it shows the efficient feature extraction ability of deep

learning model. Compared with DSCNN and BiLSTM, the proposed model has improved in all evaluation indicators, and compared with DSCNN-BiLSTM deep learning network without adding attention mechanism and optimization algorithm, the proposed model has reduced about 2.34% and 2.71% in MAPE and SMAPE, respectively. The validity of the proposed method is further verified.

**Table 4.** Prediction performance indicators of different models.

| Predictive Model | Evaluation Indicators | | | | |
|---|---|---|---|---|---|
| | MAE | MSE | RMSE | MAPE | SMAPE |
| SVR | 0.0571 | 0.0072 | 0.0850 | %6.3006 | %6.2532 |
| DSCNN | 0.0422 | 0.0026 | 0.0511 | %4.5046 | %4.6418 |
| BiLSTM | 0.0524 | 0.0041 | 0.0640 | %5.8822 | %5.6673 |
| DSCNN-BiLSTM | 0.0484 | 0.0035 | 0.0594 | %5.2893 | %5.3333 |
| Our model | 0.0269 | 0.0010 | 0.0316 | %2.9543 | %2.9597 |

### 4.4. Analysis of Results

Based on the model proposed in this paper, a product design scheme considering customer preference is designed. Five monitors of a certain brand are selected, as shown in Table 5, and the customer satisfaction curve is shown in Figure 13 as the solid red line. When other parameters remain unchanged and 3.5 inches of the five monitors are reduced, it can be preliminarily concluded that for the first four monitors, reducing some screen sizes will not have a significant impact on customer experience, but will improve the overall experience of customers. For the fifth type of 1299 yuan monitor, when the size of the monitor is reduced, the negative feelings of customers to this product will increase significantly, and the satisfaction of customers' comprehensive evaluation will decrease. As an example, this paper puts forward the method of product design follow the principle of maximizing customer satisfaction, if the value does not conform to the strategic planning of new products, the tendency of target can be set product evaluation, then gradually adjusting design parameters, and the input model to generate the corresponding image parameters, cycle to meet the design goals, to complete the new design.

**Table 5.** Experimental verification data.

| Serial Number | Brand | Price/Yuan | Resolution/px | Weight/kg | Screen Size/in | Panel Material | Customer Satisfaction |
|---|---|---|---|---|---|---|---|
| 1 | MI | 799 | 1920 × 1080 | 3.75 | 23.5 | IPS | 0.8601 |
| 2 | MI | 849 | 1920 × 1080 | 5.7 | 27 | IPS | 0.8506 |
| 3 | MI | 1299 | 2560 × 1440 | 5.7 | 27 | IPS | 0.8636 |
| 4 | MI | 749 | 1920 × 1080 | 5.3 | 23.5 | IPS | 0.8609 |
| 5 | MI | 1299 | 1920 × 1080 | 7.25 | 24.5 | IPS | 0.9527 |

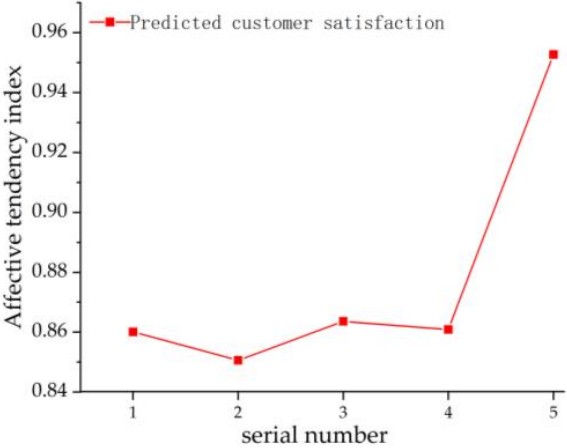

**Figure 13.** Satisfaction prediction results.



## 5. Conclusions

- This paper raises a new DSCNN-BiLSTM prediction model which integrates attention mechanism with genetic algorithm optimization, and predicts customer satisfaction degree of a product from customer online comments, thus providing scientific decision-making for enterprise production design. Firstly, a shallow feature extraction model based on deep separable convolution is adopted to fully extract the associated features between product and evaluation satisfaction data from two dimensions, effectively solving the problem of insufficient feature extraction capability of classical convolution network. Secondly, the attention mechanism is embedded for secondary depth feature extraction, and the CBAM attention distribution feature weights are used to enhance the depth feature analysis capability of BiLSTM, which improves the self-learning capability of the algorithm and effectively solves the problem of important feature loss during long-term training. Finally, the number of layers and the number of neurons per layer of the depth-separable convolutional layers, BiLSTM layers and density layers are optimised using genetic algorithms, and the model parameters are retrained and optimised by calculating the fitness function values, so as to obtain the optimal network structure of the model. Compared with traditional machine learning algorithms such as SVR and DSCNN, BiLSTM has improved the performance of our model in predicting customer affective tendency index. The performance in MSE is especially surprising to us, which is as low as 0.001.
- Meanwhile, the deep learning network optimized by genetic algorithm has good self-adaptability, self-learning ability and generalization ability. The limitation of our model performance is that when predicting new data, the model parameters need to be adjusted by genetic algorithm, which can bring high prediction accuracy, but consumes a lot of time and computing resources.
- This article takes screens as research object to analyze the fluctuation of customer satisfaction caused by the change of some important product attributes (such as price and resolution, etc.), which provides reference for enterprises in product development and design. In the future work, we plan to use the model in this paper to calculate the emotional value of texts, perform cluster analysis on online comments, and refine customers' emotional tendency towards products from different perspectives, so as to replace the comprehensive evaluation of the emotional index in this paper. In addition, the optimization design of other adjustable parameters in the model is not limited to the structure of the model to further improve the performance of the model.

**Author Contributions:** Funding acquisition, Q.Z.; Investigation, Q.Z. and J.Z.; Methodology, H.L.; Project administration, Q.Z.; Resources, Q.Z.; Supervision, Q.Z.; Writing—original draft, Q.Z. and H.L.; Writing—review & editing, Y.Z. and J.Z. All authors have read and agreed to the published version of the manuscript.

**Funding:** This research was funded by the National Key Research and Development Program, grant numbers 2018YFB1700702; the Natural Science Research Program of Anhui Agricultural University, grant numbers K2048004; the University Natural Science Research Program of Anhui Province, grant numbers KJ2021A0181; the Talent Research Program of Anhui Agricultural University, grant numbers rc482003.

**Institutional Review Board Statement:** Not applicable.

**Informed Consent Statement:** Not applicable.

**Data Availability Statement:** Data available on request due to privacy restrictions. The data presented in this study are available on request from the corresponding author. The data are not publicly available due to the privacy implications of our data.

**Conflicts of Interest:** The authors declare no conflict of interest.

## Abbreviations

| Abbreviation | Paraphrase |
| --- | --- |
| DSCNN | Depthwise Separable Convolutions |
| CBAM | Convolutional Block Attention Module |
| BiLSTM | Bidirectional Long Short-Term Memory |
| SVR | Support Vector Regression |
| QFD | Quality Function Deployment |
| HOQ | House of Quality |
| BP | Back Propagation |
| ss-rae | semi-supervised learning recursive autoencoder |
| NSGA-II | Non-inferior Sorting Genetic Algorithm II |
| SD | Semantic Differential |
| LSTM | Long Short-Term Memory |
| GA | Genetic Algorithm |
| CNN | Convolutional Neural Network |
| DWC | Depthwise Convolution |
| PWC | Pointwise Convolution |
| RNN | Recurrent Neural Network |
| CAM | Channel Attention Module |
| SAM | Spatial Attention Module |
| MLP | Multi-layer Perceptron |
| MSE | mean square error |
| MAE | mean absolute error |
| RMSE | root mean square error |
| MAPE | mean absolute percentage error |
| SMAPE | symmetric mean absolute percentage error |

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
