# Peer review of "Product Evaluation Prediction Model Based on Multi-Level Deep Feature Fusion"

_futureinternet, doi:10.3390/fi15010031_

Round 1

Reviewer 1 Report

The reviewed article is titled "Product evaluation prediction model based on multi-level deep feature fusion". It was prepared in a way typical for scientific works. The first chapter presents an introduction to predictive issues and methods of their evaluation. The parameters of the described model were presented and the literature was shown. The second section discusses the algorithm used to optimize spatiotemporal correlation. In addition, information on feature extraction was presented. The depthwise separable convolution structure was also presented. In order to make the prediction, the LSTM model is used, which has been described in section 2.1.2. Section 2.2. exposes the details of the DSCNN-BiLSTM model. Parameters allowing to reconstruct the conducted once again experiments were presented. Detailed information on individual fragments of the discussed model has been provided. This chapter also shows the course of the research process. The article also evaluates the obtained results based on commonly used measures. The article was written in the right way, the research methodology also leaves no doubt. However, I would like to ask to clarify a few things:

1. What is a role of attention mechanism in this case, please clarify in the section 2.

2. On Fig. 2. that does H mean?

3. What is the reason of using control gate in LSTM module?

4. What were the initial weights in MLP (Fig. 6), generated randomly?

5. It would be worth comparing the obtained results with those obtained in other studies.

Author Response

Dear reviewer:

Reviewer 2 Report

The major limitation of the paper is the writing style of authors especially handling of abbreviations. 

1. I recommend a table of abbreviations and their meaning be added as appendix to the paper.

line 44: delete house of quality. simply used HOQ as defined in line 40-41

line 49: what is BP?

line 51: line 51 style of abbreviations is not same as line 40-41 remove the capitalization

line 81: cbam, lstm, rnn etc not explained or defined in full

Author Response

Dear Reviewer:

Reviewer 3 Report

It is an interesting article with a clear description of the research need; however, it requires a few improvements before publication. It is advisable to improve the reproducibility of the described method in terms of the used dataset. The experiment shows promising effects of the proposed method, but it is still a few considerations that need to be addressed before the publication of the article:

-    due to the relatively large number of abbreviations used in the article, it is advisable to add a list of used abbreviations,

-   the readability of Figure 9 should be improved,

-   Line 309 - probably a repetition - The range of the emotional value is [0,1]. The range of emotion value is [0,1], where 1 represents excellent evaluation and 0 represents extreme poor,

-   Equation (15) - name error (SAMPE),

-   Figure 11 - it is necessary to unify the axes (a) or show results for different methods on a common graph,

- The transformation of the original data into the dataset used in the experiments is described too generally - it is advisable to present this process graphically,

- The size of the dataset used and the method of division into training, validation and test sets are not provided,

- Figure 12 - it is necessary to explain in more detail what both curves represent (green and red) - the descriptions of these lines should be consistent for the data from Figure 12 and Table 5 (additionally, no numerical values for the green line in Table 5),

- The obtained results were commented too generally - please add more detailed conclusions (section Conclusion),

-  It is advisable to, discuss the limitations in terms of the possibility of using the proposed method

- It is not stated how data will be made available for the reproducibility of this experiment.

Author Response

Dear reviewer:

Round 2

Reviewer 2 Report

1. Line 16: "are used to combined" please correct the sentence to "are used to combine

2. Figure 8 is incomplete. Author should show the answers or decision of the decision box: 

Example "iteration terminated? Yes or No should be shown in the diagram

3. Line 308 Support Vector Regression (SVR) 

Author Response

Dear reviewer:

We sincerely thank you for your advice. We have corrected the three problems you raised in time.